# Characterization of the Nitrate Transporter gene family and functional identification *of HvNRT2.1* in barley (*Hordeum vulgare* L.)

**Baojian Guo**[1,2,3☯], **Ying Li**[1,2,3☯], **Shuang Wang**[1,2,3☯], **Dongfang Li**[1,2,3], **Chao Lv**[1,2,3], **Rugen Xu**[1,2,3]*

**1** Jiangsu Key Laboratory of Crop Genetics and Physiology/ Key Laboratory of Plant Functional Genomics of the Ministry of Education/ Jiangsu Key Laboratory of Crop Genomics and Molecular Breeding, Agricultural College of Yangzhou University, Yangzhou, China, **2** Jiangsu Co-Innovation Center for Modern Production Technology of Grain Crops, Yangzhou University, Yangzhou, China, **3** Barley Research Institution of Yangzhou University, Yangzhou University, Yangzhou, China

☯ These authors contributed equally to this work.

* rgxu@yzu.edu.cn

**Data Availability Statement:** All relevant data are within the paper and its Supporting Information files.

**Funding:** This work was supported from the National Natural Science Foundation of China

## Abstract

Nitrogen use efficiency (NUE) is the efficiency with which plants acquire and use nitrogen. Plants have high-affinity nitrate transport systems, which involve certain nitrate transporter (NRT) genes. However, limited data are available on the contribution of the NRT2/3 gene family in barley nitrate transport. In the present study, ten putative *NRT2* and three putative *NRT3* genes were identified using bioinformatics methods. All the *HvNRT2/3* genes were located on chromosomes 3H, 5H, 6H or 7H. Remarkably, the presence of tandem repeats indicated that duplication events contributed to the expansion of the *NRT2* gene family in barley. In addition, the *HvNRT2/3* genes displayed various expression patterns at selected developmental stages and were induced in the roots by both low and high nitrogen levels. Furthermore, the overexpression of *HvNRT2.1* improved the yield related traits in *Arabidopsis*. Taken together, the data generated in the present study will be useful for genome-wide analyses to determine the precise role of the *HvNRT2/3* genes during barley development, with the ultimate goal of improving NUE and crop production.

## Introduction

Nitrogen, as a function of nutrient availability, plays an essential role in controlling plant growth and development [1]. Limited nitrogen supplies decrease crop production. However, the excessive use of nitrogen fertilizers results in severe pollution and environmental deterioration [2]. Therefore, it is important to improve nitrogen use efficiency for cereal production with a low nitrogen supply [3].

Improving nitrogen use efficiency (NUE) in plants requires a more complete understanding of the transport of $NO_3^-$ from the soil to the plant and within the plant itself. To date, two kinetically distinct nitrate uptake systems have been identified by physiological and molecular studies in plant roots, including the low-affinity transport system (LATS), which is encoded by

(31771771, 31401370 and 31571648), Natural Science Foundation of the Jiangsu Higher Education Institutions of China (17KJB210006), National Barley and Highland Barley Industrial Technology Specially Constructive Foundation of China (CARS-05), and a Project Funded by the Priority Academic Program Development of Jiangsu Higher Education Institutions. The funders had no role in study design, data collection and analysis, decision to publish, or preparation of the manuscript.

**Competing interests:** The authors have declared that no competing interests exist.

the NITRATE TRANSPORTER 1/PEPTIDE TRANSPORTER (NRT1/PTR) family (NPF), and the high-affinity transport system (HATS), which is encoded by the nitrate transporter 2 (*NRT2*) family [4–8]. The functions of some *NRT* genes have been analyzed in plants. In *Arabidopsis*, *AtNRT1.1* (*CHL1*), a dual-affinity nitrate transporter, functions as high-affinity and low-affinity nitrate transporters, when it is phosphorylated and dephosphorylated, respectively [9]. *OsNRT1.1B* (*OsNPF6.5*) is a putative homolog of *CHL1*, and it contributes to grain yield and NUE in rice [10]. The overexpression of *OsNRT1.1A* (*OsNPF6.3*) greatly improves N utilization and grain yield, as well as shortens the maturation time [11]. There is a lesser number of *NRT2* genes than *NRT1*, and they are expressed predominantly in the root [8, 12]. In rice, *OsNRT2.3* is transcribed into two spliced isoforms (*OsNRT2.3a* and *OsNRT2.3b*), and the overexpression of *OsNRT2.3b* leads to an improvement in grain yield and NUE by 40% [13]. However, when *OsNRT2.3a* expression is decreased, xylem loading of nitrate is impaired, and there is decreased plant growth at low nitrate levels [14]. This indicates that both alternative splices execute different functions in rice development.

NRT3 (also named NAR2) proteins are partner proteins that interact with most NRT2 proteins and contribute to high-affinity nitrate uptake [15]. In *Atnrt3.1* mutant plant, the expression levels of *AtNRT1.1* and *AtNRT2.1* are reduced in response to nitrate induction, and constitutive high-affinity influx and high-affinity nitrate-inducible influx are dramatically decreased [15]. Similarly, *OsNRT2.1*, *OsNRT2.2* and *OsNRT2.3a* are also synchronously suppressed in *Osnar2.1* mutant, while high- and low-affinity nitrate transport is greatly impaired. A further analysis revealed that *OsNAR2.1* not only interacts with *OsNRT2.1/OsNRT2.2* but also with *OsNRT2.3a* [16]. Thus, the *NRT2* genes combine with the *NRT3* genes to enable the plant to cope with variable nitrate supplies.

In barley, four *HvNRT2* (*HvNRT2.1-2.4*) and three HvNRT3 (*HvNAR2.1-2.3*) genes were isolated from the roots. The expression pattern of the *HvNRT*2 genes was also characterized under various nitrogen sources, and a highly specific interaction is suggested between *HvNRT2.1* and *HvNAR2.3* by using $^{15}$N-enriched nitrate uptake into Xenopus oocytes. However, until now, none of the *HvNRT2/3* gene family have been described at the barley genome level and their functions are still unclear [17–19]. In the present study, we identified *HvNRT2* and *HvNRT3* genes in barley using bioinformatics and constructed a phylogenetic tree. Then, the expression patterns of these genes in response to low and high nitrate levels were analyzed. In addition, the *HvNRT2.1* gene was overexpressed to determine its function in *Arabidopsis*. These results will be useful in further investigations into the functions of the *NRT* family in high-affinity nitrate uptake in plants.

## Materials and methods

### Plant materials

Barley cv. Morex was used in the present study. The seeds were surface-sterilized and germinated on wet filter paper at 25˚C for 3 days. The germinated seeds were transferred into 60-well plastic containers (25 L) with aerated and nitrogen-free one-tenth-strength modified Johnson's solution for 4 days [20]. The plants were then supplied with 0.2 mM $NO_3^-$ (Low nitrogen, LN) or 5 mM $NO_3^-$ (High nitrogen, HN). The plants were grown in a growth chamber at 25˚C/22˚C (day/night) under a 16-h/8-h light/dark cycle and 70% relative humidity. At 0 h, 1 h, 3 h, 6 h, 12 h, 24 h, 48 h, and 72 h after the treatment, the total roots were harvested from ten plants and were immediately frozen in liquid nitrogen for RNA extraction, with three biological replicates at each time point. For each replicate, the RNA from four plants of each genotype was analyzed. The *Arabidopsis thaliana* Columbia-0 (Col-0) ecotype was used as the wild type. The plants were grown in a greenhouse in soil at 20˚C under long-day conditions

(16 h light/8 h dark). For the *in vitro* seedling assays, the seeds were surface sterilized and cold treated at 4˚C for 3 days in the dark and were then exposed to white light. The plants were grown at 20˚C on horizontal or vertical plates containing Murashige and Skoog (MS) medium, 3% sucrose, and 0.9% agarose (Merck).

## Sequence database searches

The amino acid sequences of the *NRT2* and *NRT3* genes generated from *Arabidopsis* [15, 21], rice [22], and maize [23, 24] were used as query sequences. The barley sequence data were sourced from the Morex (http://webblast.ipk-gatersleben.de/barley/) [25], Gramene (http://ensembl.gramene.org/Hordeum_vulgare/Info/Index), and NCBI databases (http://www.ncbi.nlm.nih.gov/). BLAST programs (TBLASTN and BLASTN) were available for the IPK barley genome database and the NCBI barley EST database. Multiple database searches were performed to collect all members of the barley *NRT2* and *NRT3* genes. Domain searches (PF07690.14 and PF16974.3) were performed using SignalP (http://www.cbs.dtu.dk/services/SignalP/) [26], HMMSCAN (https://www.ebi.ac.uk/Tools/hmmer/search/hmmscan) [27], and SMART (Simple Modular Architecture Research Tool: http://smart.embl-heidelberg.de/) [28], with the default cutoff parameters. The isoelectric points and protein molecular weights were obtained with the help of the proteomics and sequence analysis tools on the ExPASy proteomics server (http://expasy.org/) [29]. The names of the *HvNRT2* genes were designated according to their ascending order in barley chromosomes. However, the names of the *HvNRT3* genes were given according to the definition provided by Tong et al. [30].

## Chromosomal location, gene structure, and duplication events of the *NRT2/3* genes

The chromosomal locations were retrieved from the IPK database (http://webblast.ipk-gatersleben.de/barley/). All the genes were mapped to the chromosomes with MapDraw software [31]. The exon/intron structures were constructed using GSDS (http://gsds.cbi.pku.edu.cn/) [32]. Tandem duplication genes were identified manually if they were within the 10 predicted genes or within 30 kb of each other on the physical barley map [33]. Segmental duplications were identified by BLASTP in ten predicted proteins upstream and downstream of each of the *HvNRT2/3* genes [34].

## Phylogenetic tree analysis

Full-length amino acid sequences of the *HvNRT2/3* genes identified in barley were aligned using the Clustal X 1.83 program with the default pairwise and multiple alignment parameters. The phylogenetic tree was constructed based on this alignment using the neighbor joining (NJ) method in MEGA version 6 with the following parameters: Poisson correction; pairwise deletion; uniform rates; and bootstrap (1000 replicates) [35]. Conserved motifs were investigated by multiple alignment analyses using MEME version 3.0 [36].

## *NRT2* and *NRT3* gene expression analyses

Gene expression data from eight tissues of the cultivar 'Morex' were obtained from the barley genome database (http://apex.ipk-gatersleben.de/apex/f?p=284:10:6281639160219::NO). Deep RNA sequencing (RNA-seq) was carried out on fifteen Morex tissues from almost all stages of the barley life cycle, which was comprised 4-day embryos derived from a germinated seed, the roots and shoots from a 10 cm seedling (10 cm shoot stage), inflorescences (5 cm and 1-1.5 cm), developing grains at 5 and 15 days after pollination (DAP), an etiolated seedling (10

DAP), the epidermal strips and roots (28 DAP), rachis of inflorescences (35 DAP), lemma, lod-icule, and palea (42 DAP), developing tillers at the 3rd internode (42 DAP), and senescing leaves (56 DAP) [25]. The expression patterns are presented as heat maps in green/yellow/red coding, which reflects the FPKM (Fragments Per Kilobase of transcript per Million mapped reads), with red indicating a high expression level, yellow indicating a moderate expression level, and green indicating a low expression level.

## Isolation of total RNA and quantitative real-time PCR

Total RNA was isolated using an RNA extraction kit (TRIzol reagent, Invitrogen, USA), and the isolated RNA was incubated with RNase-free DNase I (TaKaRa, Japan) to remove any con-taminating DNA. The RNA quality and yield were analyzed by agarose gel electrophoresis and a NanoDrop 1000 Spectrophotometer V 3.7. First strand cDNA was generated from 2 μg of total RNA with M-MLV reverse transcriptase (TaKaRa, Japan) using random primers. The specific primers used for the quantitative real-time PCR analysis are listed in S1 Table. The reactions were carried out in 20 μl reaction systems containing 10 mM Tris-HCl (pH 8.5), 50 mM KCl, 2 mM $MgCl_2$, 0.4 μl DMSO, 200 mM dNTPs, 10 pmol specific PCR primers, 1 U Taq DNA polymerase, and 0.5 μl SYBR GREEN I fluorescence dye. Quantitative real-time PCR was performed using a ViiA™ 7 Real-Time PCR System(Applied Biosystems, USA). The running protocol was as follows: 94˚C for 3 min, followed by 40 cycles at 94˚C for 30 s; 58˚C for 30 s; 72˚C for 30 s; and a final extension of 72˚C for 5 min. The amplification of *HvActin* (Accession number HORVU1Hr1G002840) was employed as an internal standard. All the reactions were run in triplicate. The *Ct* values were determined by ViiA™ 7 software using the default settings. The relative expression levels of the target genes were determined using the $2^{-\Delta\Delta Ct}$ method [37]. For each sample, the PCR was performed with three biological replicates.

## Generation of *Arabidopsis* transgenic plants

The full-length coding sequence of *HvNRT2.1* was cloned into the Gateway entry vector pDNOR221 (S1 Table). After confirmation by DNA sequencing, the *HvNRT2.1* fragments were recombined into the pB2GW7 destination vector, which expresses under the control of the CaMV35S promoter. The resulting construct was introduced into the *Agrobacterium tume-faciens* strain GV3101 and was used to transform the *Arabidopsis* plants using the floral dip method. The transgenic plants were selected by herbicide treatment, and three representative T3 homozygous lines were obtained for the phenotypic analysis. The statistical analysis of the differences between the wide type and transgenic plants was performed by using a Student's *t*-test.

## Silique and seed size measurements

Mature siliques were harvested, and their lengths were recorded using vernier calipers. The seed size measurement was performed as described previously [38]. Briefly, the siliques were harvested once they were dry. The seeds were allowed to dry completely in centrifuge tubes for at least seven days before the measurement. Next, the seeds were spread and taped onto a piece of white paper, so that none of the seeds were touching each other. Using a flatbed scanner (Canon CanoScan 9000F), images of each accession was obtained at a resolution of 300 dpi with transmitted light. The "particle analysis" feature of ImageJ software was used to measure the seed length, width, and area. Three biological replicates were undertaken with 10 plants per transformed plant.

### Nitrate and nitrogen content measurements

To determine the tissue nitrate content at maturity, Col-0 and transgenic plants were weighed and boiled in distilled water for 5 min, and the nitrate content was determined by the Cataldo method [39]. Whole plants were dried to a constant weight at 80°C and were ground in a Cyclotec 1093 sample mill (Hoganas City, Sweden) before being sieved through a 0.5 mm screen. The total nitrogen content in the plants was quantified according to the Kjeldahl method by using FOSS Kjeltec TM 2300 (Foss Analytical AB, Sweden) [40]. The statistical analysis of the differences in aerial part traits between the Col-0 and transgenic plants was performed using a Student's *t-test*.

## Results

### Identification of *NRT2/3* genes in the barley genome

A total of ten putative *HvNRT2* and three putative *HvNRT3* genes from the entire barley genome were identified (Table 1). The *HvNRT2* genes encoded proteins ranging from 483 (HvNRT2.10) to 514 (HvNRT2.2) amino acids, with protein masses that ranged from 50.40 kD to 55.38 kD and protein *pI* values ranging from 7.01 (HvNRT2.10) to 8.98 (HvNRT2.1) (Table 1). However, the *HvNRT3* genes encoded low molecular weight proteins compared to the *HvNRT2* genes, with the protein masses ranging from 21.13 kD to 21.27 kD. These were also basic proteins (the *pI* values ranged from 9.00 to 9.23). A further analysis revealed that *HvNRT3.2* was located on the end of 5HL, *HvNRT2.2-9* was on the end of 6HS, *HvNRT3.1* on the centromeric regions of 6H, and the remaining genes were distributed on the long arm of chromosomes 3, 6, and 7 (Fig 1). Remarkably, the duplication event analysis indicated that 8 (8/10, 80%) of the *HvNRT2* genes were tandem repeated (Fig 1). The tandem duplicated genes contained two clusters, and each cluster contained four genes (*HvNRT2.2-5* and *HvNRT2.6-9* gene pairs). The gene structural analyses showed that the *HvNRT2* genes contained one exon. However, the *HvNRT3* genes shared three exons (S1 Fig).

### Conserved amino acid sequence and phylogenetic tree for the HvNRT2/3 proteins in plants

Bioinformatic methods were employed to analyze the HvNRT2 and HvNRT3 proteins in the present study, revealing that an MFS domain and an NAR domain were present in the

**Table 1. The information of *HvNRT2/3* genes in barley.**

| Gene Name | Gene ID | Chr. | Physical Position on Barley Genome (start position (bp)-end position (bp)) | Coding Sequence Length (bp) | Amino Acid Length (aa) | *Mass* (Da) | *pI* |
|---|---|---|---|---|---|---|---|
| *HvNRT2.1* | HORVU3Hr1G066090.1 | 3 | 503310429-503312717 | 1542 | 514 | 55376.14 | 8.98 |
| *HvNRT2.2* | HORVU6Hr1G005570.1 | 6 | 12363356-12364512 | 1521 | 507 | 54703.55 | 8.39 |
| *HvNRT2.3* | HORVU6Hr1G005580.1 | 6 | 12371132-12373090 | 1521 | 507 | 54673.52 | 8.39 |
| *HvNRT2.4* | HORVU6Hr1G005590.1 | 6 | 12378589-12380579 | 1521 | 507 | 54627.44 | 8.39 |
| *HvNRT2.5* | HORVU6Hr1G005600.2 | 6 | 12385842-12387748 | 1527 | 509 | 54974.94 | 8.53 |
| *HvNRT2.6* | HORVU6Hr1G005720.1 | 6 | 12654106-12655479 | 1524 | 508 | 54658.86 | 8.67 |
| *HvNRT2.7* | HORVU6Hr1G005770.1 | 6 | 12754340-12756226 | 1527 | 509 | 54553.54 | 8.14 |
| *HvNRT2.8* | HORVU6Hr1G005780.1 | 6 | 12765305-12766893 | 1521 | 507 | 54367.15 | 7.9 |
| *HvNRT2.9* | HORVU6Hr1G005930.1 | 6 | 13075246-13077142 | 1524 | 508 | 54636.74 | 8.17 |
| *HvNRT2.10* | HORVU7Hr1G098550.4 | 7 | 598494068-598496471 | 1449 | 483 | 50402.35 | 7.01 |
| *HvNAR3.1* | MLOC_3053.1 | 6 | 268,053,711-268,054,907 | 591 | 197 | 21129.3 | 9.11 |
| *HvNAR3.2* | HORVU5Hr1G115500.3 | 5 | 646684461-646686179 | 777 | 259 | 21268.37 | 9 |
| *HvNAR3.3* | HORVU6Hr1G053710.1 | 6 | 336809019-336810586 | 594 | 198 | 21137.32 | 9.23 |

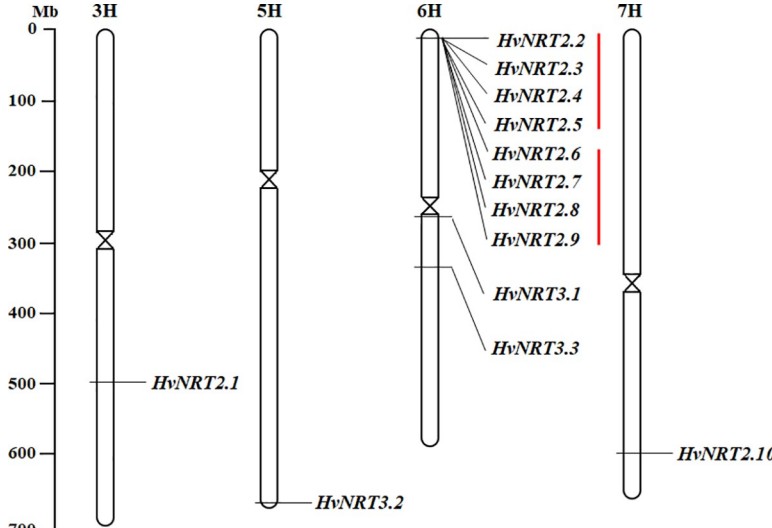

**Fig 1. The chromosomal location of the *HvNRT2/3* genes in the barley physical map.** The red lines represent tandemly duplicated gene pairs.

HvNRT2 and HvNRT3 proteins, respectively (S2 Fig). The alignment and comparison of the HvNRT2 full length protein sequences illustrated that the proteins had ten (HvNRT2.10), eleven (HvNRT2.1/6/8), or twelve (HvNRT2.2/3/4/5/7/9) (S2 Fig) transmembrane domains, while only one transmembrane region was found in the HvNRT3 proteins.

According to the phylogenetic tree of the NRT2 and NRT3 proteins in barley, rice, maize, and *Arabidopsis*, the NRT2 proteins could clearly be divided into three distinct clusters (I, II, and III). A total of 18 *NRT2* genes belonged to cluster I, including five *NRT2* genes derived from dicotyledonous *Arabidopsis* and 13 *NRT2* genes from monocotyledonous plants (8, 2, and 3 genes from barley, rice, and maize, respectively). *AtNRT2.5* and *AtNRT2.7* represented clusters II and III, containing four and three genes, according to the phylogenetic tree, respectively (Fig 2A). In the NRT3 phylogenetic tree, the proteins derived from the dicotyledonous and monocotyledonous plants clustered together (Fig 2B), indicating an evolutionary dichotomy of the *NRT2/3* genes between the dicot and monocot plants. In addition, the number of *NRT2* and *NRT3* genes in barley was larger than in other plants.

## *HvNRT2/3* gene expression patterns at different developmental stages

An exploration of the expression patterns of genes contributes to our understanding of their molecular function [41]. Thus, we searched the deep RNA sequencing (RNA-seq) data from the cultivar 'Morex' [25, 42]. A heat map of the FPKM signal values for the 13 *HvNRT2/3* genes in the selected Morex tissues is presented in Fig 3. All of the *HvNRT2/3* genes displayed relatively high expression levels in the 4-day embryos dissected from germinating seeds (except *HvNRT2.10*) and in roots (10 cm shoot stage and 28 DAP). Conversely, a limited expression of the *HvNRT2/3* genes was observed in the developing inflorescences (0.5-1.5 cm) and grains (5 and 15 DAP). In addition, *HvNRT2.10* had a higher expression level than other genes in the shoots (10 cm shoot stage), while *HvNRT3.2* had a relatively high expression level in the senescing leaves. Remarkably, *HvNRT3.2* showed an exceptionally high expression level compared to the other genes in the developing tillers (the third internode at 42 DAP), etiolated seedlings (10 DAP), lemmas (42 DAP), lodicules (42 DAP), palea (42 DAP), epidermis (28 DAP), and

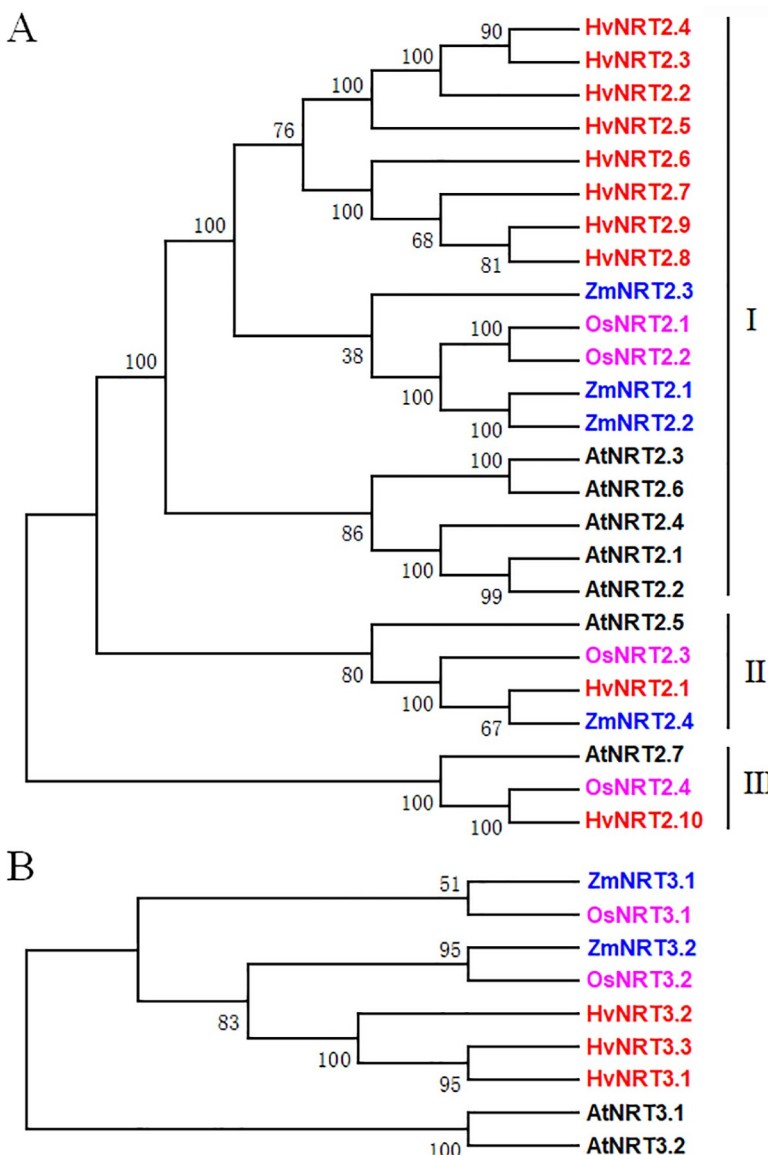

**Fig 2. Phylogenetic analysis of the NRT2/3 proteins in *Arabidopsis*, maize, rice, and barley.** A, the phylogenetic analysis of the NRT2 genes; B, the phylogenetic analysis of the NRT3 genes. *NRT2* gene sequence of *Arabidopsis*, maize, and rice generated from the report by Plett et al. [20].

rachises (35 DAP). This expression pattern analysis may contribute to our understanding of the function of the *HvNRT2/3* genes in barley.

## *HvNRT2/3* gene expression analysis under low and high nitrogen levels

The expression pattern of the *HvNRT2* and *HvNRT3* genes in barley roots was investigated using quantitative real-time PCR. As shown in Fig 4, when the barley roots were exposed to low (0.2 mM) and high (5 mM) nitrogen levels, the number of transcripts of seven *HvNRT2* and two *HvNRT3* genes increased, with peaks at 1 h (*HvNRT2.1/10* genes), 3 h (*HvNRT2.2/5* genes), and 6 h (*HvNRT2.3/4/9* and *HvNRT3.1/2* genes), before gradually decreasing. However, the mRNA levels of *HvNRT2.6/7/8* rapidly increased and peaked at 6 h under the low

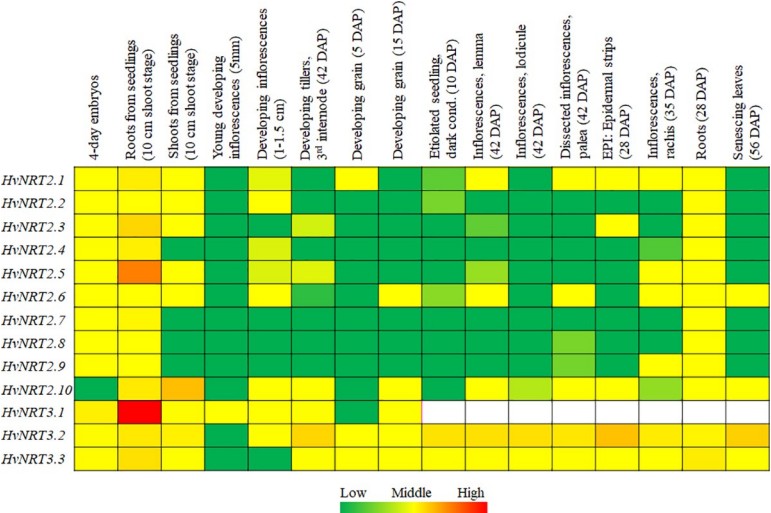

**Fig 3. Expression profile analysis of the *HvNRT2/3* genes in barley.** The expression patterns are presented as heat maps in green/yellow/red coding, which reflects the FPKM, with red indicating high expression levels, yellow indicating moderate expression levels, and green indicating low expression levels. The data on the expression of the *HvNRT3.1* gene were only gathered from eight selected tissues [39]. The default values are shown in white.

nitrogen conditions and at 72 h under the high nitrogen conditions. In addition, the expression of *HvNRT3.3* displayed two peaks at 3 h and 24 h and responded more quickly to high rather than low nitrogen conditions.

## Morphological characterization of the *HvNRT2.1* overexpression plants

In order to investigate the functional roles of the barley *NRT* genes, *HvNRT2.1*, which has a high homology to *OsNRT2.3*, and was overexpressed in *Arabidopsis*. In the transgenic lines, *HvNRT2.1* was introduced into the genomic DNA and was expressed at the transcription level in three independent overexpression plants (S3 Fig). Phenotypic changes were observed between the Col-0 and overexpression plants at the mature stage (Fig 5). There were no obvious differences in the plant height, dry weight, and number of siliques per plant between the Col-0 and transgenic lines (Fig 6A, 6B and 6D). However, 7.9%, 18.5%, and 23.3% increments in the length of siliques, yield per plant and grain number per siliques, respectively, were observed in the transgenic lines compared with Col-0 (Fig 6C, 6E and 6F). The size of the seeds, including the seed length, width, and thousand grain weight, was higher in the three independent *HvNRT2.1* transgenic lines compared to the wild type (Fig 6G, 6H and 6I). The seed length and width significantly increased by 11.5% and 24.5% compared to the wild type, respectively. The thousand grain weight dramatically increased by 37.0%, suggesting that the barley *HvNRT2.1* gene positively regulated seed development. In addition, the seeds from the overexpression plant accumulated significantly more nitrate and nitrogen, and these increased by 47.2% and 41.4%, respectively, compared with the wide type for the mature plants (Fig 7).

## Discussion

Both NRT2 and NRT3 proteins are involved in high-affinity nitrate uptake in plants [15]. Due to the sequencing of plant genomes, the *NRT2/3* genes have been widely analyzed in plants. For example, a total of 4, 7, and 4 *NRT2* genes have been analyzed in rice, *Arabidopsis*, and maize, respectively, as well as two *NRT3* genes [8, 15, 23–24, 43]. Although several *HvNRT2*

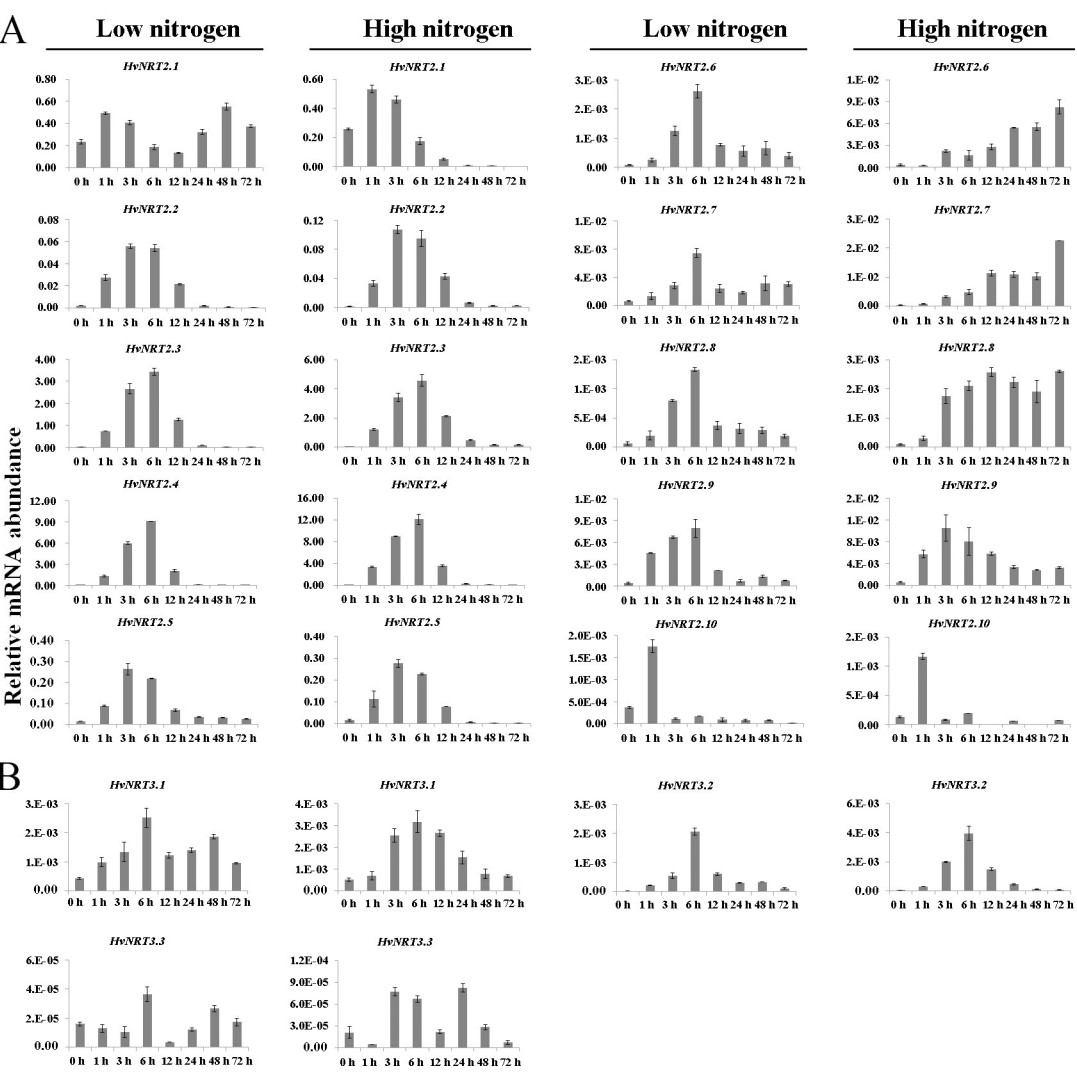

**Fig 4. Expression patterns of the *HvNRT2/3* genes in the roots pretreated with low and high nitrogen levels.** A, expression patterns of the *HvNRT2* genes; B, expression patterns of the *HvNRT3* genes.

and *HvNRT3* genes have been cloned in barley [30, 44], there is no information about barley *HvNRT2/3* genes at the genome level. The present study identified ten putative *NRT2* and three putative *NRT3* genes that encode components of the high-affinity nitrate transport system (HATS) in barley. The number of *NRT2/3* genes in barley is greater than those in rice, *Arabidopsis*, and maize. A duplication event analysis indicated that tandem duplication played an important role in the expansion of the *NRT2* genes and may have contributed to the difference in the number of *NRT* genes in barley, rice, *Arabidopsis*, and maize [45]. In addition, SignalP and SMART predicted that three HvNRT3 proteins had signal peptides with 21 (HvNRT3.2/3) and 25 (HvNRT3.1) amino acids (S2 Table), which was similar to that of the *AtNRT3* genes in *Arabidopsis*.

Phylogenetic analyses and evolutionary relationships are used to investigate the functions of genes in plants [46]. In rice, *OsNRT2.1* and *OsNRT2.2* share identical amino acid sequences and are clustered together with other *NRT2* genes. Rice is a monocotyledon plant known to have undergone tandem duplication [8]. Interestingly, similar events were detected at the end

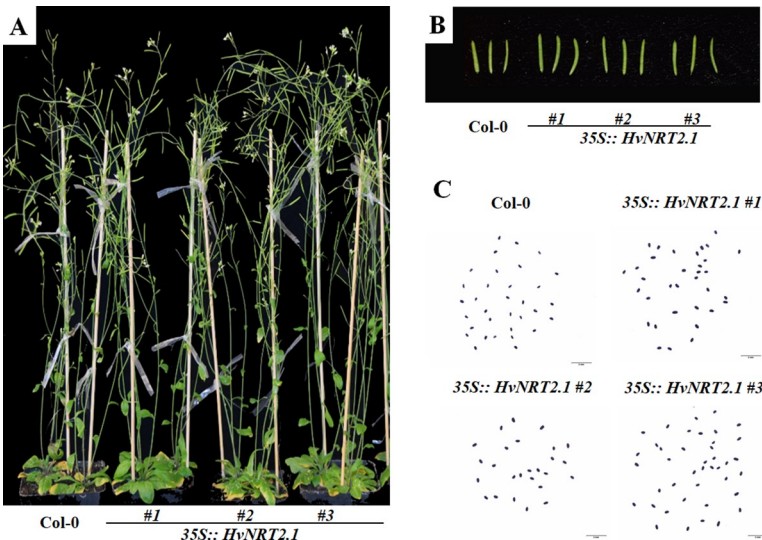

**Fig 5. Phenotype of the overexpression and Col-0 plants.** A, Whole plant; B, Silique; C, Seed.

of 6HS, where the *HvNRT2.2/3/4/5* and *HvNRT2.6/7/8/9* genes were clustered. Both HvNRT2.1 and HvNRT2.10 were closely related to OsNRT2.3 and OsNRT2.4 based on the phylogenetic tree. It is worth noting that barley has one exon in common with rice (except OsNRT2.4) but not *Arabidopsis*, suggesting that these *NRT2* genes diverged before the mono-cotyledon-dicotyledon separation in higher plants [8, 12]. The structure of the *NRT3* genes with two exons is conserved in plants, while divergence is observed in the protein homology between monocotyledon and dicotyledon plants [8, 15].

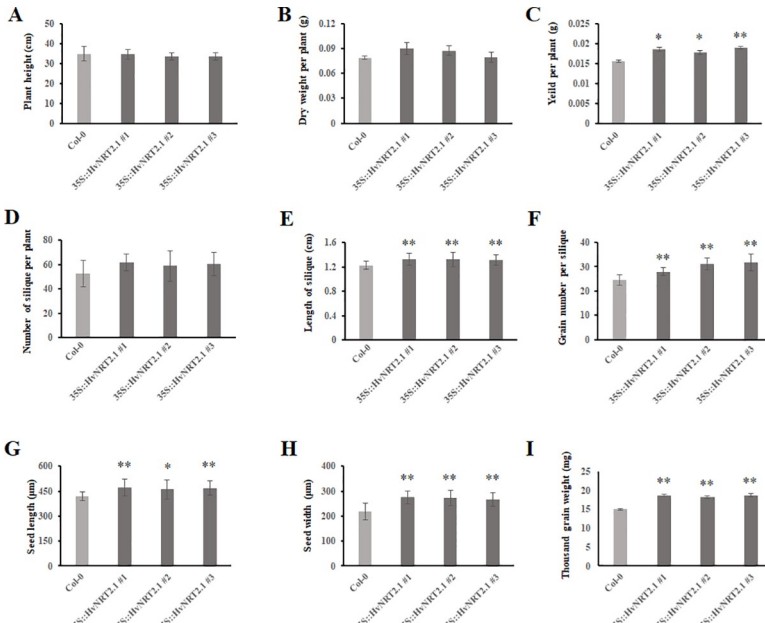

**Fig 6. Phenotype analysis of the Col-0 and overexpression plants.** A, plant height; B, dry weight; C, yield per plant; D, number of silique per plant; E, length of silique; F, grain number per plant; G, seed length; H, seed width; I, thousand grain weight. $*p < 0.05$; $**p < 0.01$.

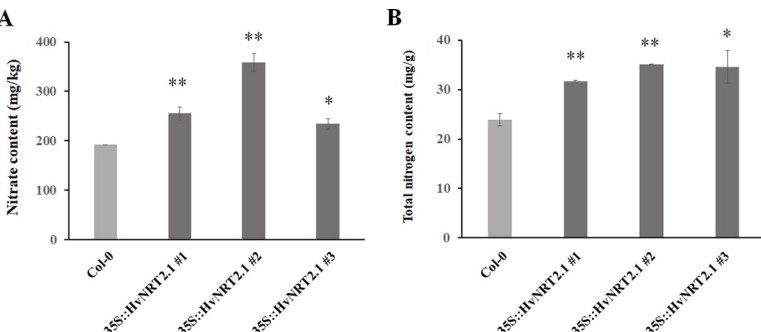

**Fig 7. Nitrate and nitrogen content in the Col-0 and overexpression seeds.** A, nitrate content; B, nitrogen content.
$*p < 0.05$; $**p < 0.01$.

Expression analyses of organs is employed to predict the potential role of specific genes in plant growth and development, as well as their response to different environments [47]. *NRT2* genes, as members of the high-affinity transport system, combined with *NRT3* genes, were detected in the roots with high abundance levels, indicating that both gene families are responsible for induction either at low or high nitrate levels in the root tissue [8, 12, 15]. All of the thirteen genes investigated had high expression levels in the root tissue and displayed obvious changes following low and high nitrogen treatments. Remarkably, *HvNRT2.10* is a homologous of *AtNRT2.7* and *OsNRT2.4*, with 45% and 51% amino acid identity, respectively. It has a similar pattern of expression compared to its homologs, with higher expression levels in the shoots than the roots, and is also relatively highly expressed in seeds, which is consistent with the *AtNRT2.7* gene [48, 49]. In rice, a decreased expression of *OsNRT2.3a* is related to the accumulation of significantly higher levels of nitrate and total nitrogen in the root and lower levels in the shoot [14]. However, enhancing the expression of *OsNRT2.3b* improves the growth, yield, and NUE in rice [13]. The HvNRT2.1 protein had a similar amino acid length and showed 83.3% identity with *OsNRT2.3a*, but only 78.4% identity with *OsNRT2.3b* genes, indicating that *HvNRT2.1* plays an important role in long-distance nitrate transport, from the roots to shoots, under low nitrate supply levels in barley. These results will be useful in further investigations of *HvNRT2.1* and *HvNRT2.10* in barley.

*NRT3* genes encode a partner protein that interacts with some members of *NRT2* in plants [16, 50]. For example, AtNRT2.1 combined with AtNRT3.1 forms a 150-kDa plasma membrane complex. Both genes were thought to constitute the high-affinity nitrate transporter of *Arabidopsis* roots [50]. *OsNRT3.1 (OsNAR2.1)* was induced by low and high nitrogen levels, and *in vitro* experiments demonstrated that OsNAR2.1 interacts with OsNRT2.1 and OsNRT2.2, as well as OsNRT2.3a [16]. Remarkably, eight *HvNRT2* genes and two *HvNRT3* genes displayed co-expression patterns under low nitrogen conditions. Therefore, the interactions between the *HvNRT2* and *HvNRT3* genes should be further investigated to clarify the functions of the NRT genes in barley.

Several *NRT2* genes have been characterized in plants. For example, an *Atnrt2.1-1* mutant was specifically deficient in HATS, and a further analysis showed that when the ATNRT2.1 protein combined with ATNAR2.1, it forms a complex that transports nitrate efficiently [15, 51, 52]. *ATNRT2.7*, which had a high expression level in the dry seeds, was localized to the vacuolar membrane and displayed a positive regulation of nitrate content in the seeds. In the present study, the overexpression of *HvNRT2.1* led to high nitrate and nitrogen contents and improved the yield-related traits of *Arabidopsis*. However, both a biomass and plant height

increase were not observed in the overexpression plants, indicating that the *HvNRT2.1* gene plays a specific role in nitrate and total nitrogen accumulation in plants.

## Conclusion

In summary, ten putative *NRT2* and three putative *NRT3* genes were identified from barley sequencing using bioinformatic methods. A duplication event analysis indicated that tandem repeats contributed to the expansion of the *NRT2* gene family in barley. A phylogenetic tree revealed that the NRT2/3 proteins displayed a clear divergence between the monocot and dicot plants. In addition, the *HvNRT2/3* genes displayed various expression patterns at selected development stages and were induced in the roots under low and high nitrogen levels. Remarkably, *HvNRT2.1*, as a homologous gene of *OsNRT2.3*, significantly improved the yield-related traits in *Arabidopsis*. Thus, the data generated in the present study will be powerful for genome-wide analyses to determine the precise role of the *HvNRT2/3* genes during barley development, with the ultimate goal of improving NUE and crop production.

## Supporting information

**S1 Fig. Gene structure analysis of the *HvNRT2/3* genes.**
(TIF)

**S2 Fig. Alignments of the HvNRT2/3 proteins.** Amino acid sequence alignments of the HvNRT2 (A) and HvNRT3 (B) proteins. The conserved transmembrane sequence regions are indicated by the black lines above the sequences. The red frame indicates the MFS and NAR domains in the HvNRT2 and HvNRT3 proteins, respectively.
(TIF)

**S3 Fig. PCR analysis of the overexpression and wild-type (Col-0) plants.** A, PCR analysis at the genomic level; B, RT-PCR analysis at the transcription level.
(TIF)

**S1 Table. Primers used in this study.**
(XLSX)

**S2 Table. Summary of the functional domains present in the NRT proteins.**
(XLS)

## Author Contributions

**Data curation:** Chao Lv.

**Formal analysis:** Chao Lv.

**Investigation:** Ying Li, Dongfang Li.

**Methodology:** Ying Li, Dongfang Li.

**Project administration:** Baojian Guo, Ying Li.

**Resources:** Dongfang Li.

**Software:** Shuang Wang.

**Validation:** Shuang Wang.

**Writing – original draft:** Baojian Guo, Rugen Xu.

**Writing – review & editing:** Baojian Guo, Rugen Xu.

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
