## [Decision Letter · Decision Letter 0]

31 Jan 2020

PONE-D-19-33678

Characterization of the Nitrate Transporter gene family and functional identification of HvNRT2.1 in barley (Hordeum vulgare L.)

PLOS ONE

Dear Dr. Xu,

Thank you for submitting your manuscript to PLOS ONE. After careful consideration, we feel that it has merit but does not fully meet PLOS ONE’s publication criteria as it currently stands. Therefore, we invite you to submit a revised version of the manuscript that addresses the points raised during the review process.

We would appreciate receiving your revised manuscript by Mar 16 2020 11:59PM. To enhance the reproducibility of your results, we recommend that if applicable you deposit your laboratory protocols in protocols.io, where a protocol can be assigned its own identifier (DOI) such that it can be cited independently in the future. For instructions see: http://journals.plos.org/plosone/s/submission-guidelines#loc-laboratory-protocols

We look forward to receiving your revised manuscript.

Kind regards,

Guoping Zhang

Academic Editor

PLOS ONE

2.Thank you for stating the following in the Acknowledgments Section of your manuscript:

"This research was supported from the National 356 Natural Science Foundation of China (31771771, 31401370 and 31571648), Natural Science Foundation of the Jiangsu Higher Education Institutions of China (17KJB210006), National Barley and Highland Barley Industrial Technology Specially Constructive Foundation of China (CARS-05), and a Project Funded by the Priority Academic Program Development of Jiangsu Higher Education Institutions."

 "NO - Include this sentence at the end of your statement: The funders had no role in study design, data collection and analysis, decision to publish, or preparation of the manuscript."

Reviewers' comments:

Reviewer's Responses to Questions

**Comments to the Author**

1. Is the manuscript technically sound, and do the data support the conclusions?

Reviewer #1: Yes

Reviewer #2: Yes

2. Has the statistical analysis been performed appropriately and rigorously? 

Reviewer #1: Yes

Reviewer #2: Yes

3. Have the authors made all data underlying the findings in their manuscript fully available?

Reviewer #1: Yes

Reviewer #2: Yes

4. Is the manuscript presented in an intelligible fashion and written in standard English?

Reviewer #1: Yes

Reviewer #2: No

5. Review Comments to the Author

Reviewer #1: This manuscript described a genome-wide characterization of high-affinity nitrate transporter NRT2/3 family members by taking advantage of the publically-released barley reference sequence. The gene duplication, transcriptional pattern and chromosomal assignment as well as the phylogenetic analysis were further represented. One NRT2-type gene HvNRT2.1 was over-expressing in transgenic Arabidopsis plants that showed the increases on the parameters of grains, as well as the accumulation of nitrate and total nitrogen uptake in plant tissues.

1, A short introduction referring to the current understanding of individual HvNRT2/3 gene in cultivated barley is suggested.

2, Fig. 7 showing the increases of nitrate and total nitrogen content should be described in Results. This might be important in order to support the story of this study.

3, Use Arabidopsis thaliana or A. thaliana. Refer the variety “Col-0” in Materials.

4, Fig. 4, statistical analysis is recommended (e.g. Tukey’s HSD).

5, The language editing is suggested.

Reviewer #2: The authors mapped the NRT2/3 gene family in the barley genome and analysed their expression patterns under N treatments. Besides, one member from the family HvNRT2.1 was exotically expressed in Arabidopsis for function characterisation. The work provides new insights into the barley NUE research and I have the following concerns before it could be accepted for publishing.

Major issues

1. The English language needs professional proofreading as there are massive typos and grammar errors in the main document.

2. Why the HvNRT2.1 but not other NRT transporters were specifically investigated in Arabidopsis?

3. Line 223, information missing in the sentence

4. Line 256, peaks at ??

5. Line 261, how much similar are they (% identity)?

6. Line 264, was Arabidopsis ecotype Col-0 used? Should not be in italic.

7. Line 267, how much increments?

8. Line 278, two NRT3 from which species?

9. Line 282, what does “components” mean?

10. Line 307-309, please specify the details in the changes.

11. Line 313-318, state one or two functions in rice so that it can be more evidently related to barley.

12. Line 335, the statement has been repeated a couple of times in the ms.

13. Line 338, increase of biomass and plant height in Arabidopsis or ?? If so could be rephrased as “In the present study, overexpression of HvNRT2.1 led to high nitrate and nitrogen content, increased biomass and plant height and improved the yield-related traits in Arabidopsis indicating that….

14. The authors should provide either qualitative or quantitive information regarding the HvNRT2.1 transcript levels in transgenic Arabidopsis lines.

15. HvNRT2.1 expression in barley responded to different N levels so why not investigate the growth performance of transgenic Arabidopsis lines under N treatments?

Minor issues

1. Line 20, rectify the grammar issue

2. Line 23, in barley “nitrate transport”

3. Line 25, “chromosomes”

4. Line 29, replace “notably” with “furthermore”

5. Line 55, should be “they are” but not “it is”

6. Line 61, under low N condition?

7. Line 86-89, confusing with the experimental layout. Indeed how many plants were sampled and used for RNA extraction?

8. Line 137, “0.5 cm” would be better.

9. Line 224, according “to”

10. Line 228, “clustered”

11. Line 320, “encode”

6. PLOS authors have the option to publish the peer review history of their article (what does this mean?). If published, this will include your full peer review and any attached files.

Reviewer #1: No

Reviewer #2: Yes: Yong Han

---

## [Author Response · Author response to Decision Letter 0]

10 Mar 2020

Responses to reviewer 1:

Comment 1: A short introduction referring to the current understanding of individual HvNRT2/3 gene in cultivated barley is suggested.

Response 1: According to the reviewer’s suggestion, we have added the sentence in the revised manuscript. In barley, four HvNRT2 (HvNRT2.1-2.4) and three HvNRT3 (HvNAR2.1-2.3) genes were isolated from the roots. The expression pattern of the HvNRT2 genes was also characterized under various nitrogen sources, and a highly specific interaction is suggested between HvNRT2.1 and HvNAR2.3 by using 15N-enriched nitrate uptake into Xenopus oocytes. However, until now, none of the HvNRT2/3 gene family have been described at the barley genome level and their functions are still unclear. 

Comment 2: Fig. 7 showing the increases of nitrate and total nitrogen content should be described in Results. This might be important in order to support the story of this study.

Response 2: We agree with this suggestion and have added the result in the “Morphological characterization of the HvNRT2.1 gene overexpression plants” paragraph as follows: In addition, seeds of overexpression plant accumulated significantly more nitrate and nitrogen, and increased by 47.2% and 41.4% compared with wide type in mature plants (Fig 7).

Comment 3: Use Arabidopsis thaliana or A. thaliana. Refer the variety “Col-0” in Materials.

Response 3: Thank the reviewer for this content. The Arabidopsis thaliana Columbia-0 (Col-0) ecotype was used as in materials. We have revised this in the new manuscript (see “Plant materials” part)

Comment 4: Fig. 4, statistical analysis is recommended (e.g. Tukey’s HSD).

Response: Thank the reviewer’s suggestion, statistical analysis of the differences between wide type (Col-0) and transgenic plants was performed by using Student’s t-test. We have added the result of Student’s t-test in Fig 6 and Fig 7.

Comment 5: The language editing is suggested.

Response: The statements have been corrected. We will be happy to edit the MS further base on helpful comments from reviewers.

Responses to reviewer 2:

Major issues

Comment 1: The English language needs professional proofreading as there are massive typos and grammar errors in the main document.

Response 1: The statements have been corrected. We really hope the flow and language level have been improved.

Comment 2: Why the HvNRT2.1 but not other NRT transporters were specifically investigated in Arabidopsis?

Response 2: HvNRT2.1 gene was the first to respond to low nitrogen with peaks at 1 h, which has high homology to OsNRT2.3 (a gene derived from rice which has been contribute to improvement grain yield and NUE by 40%). Therefore, HvNRT2.1 was specifically investigated in Arabidopsis.

Comment 3: Line 223, information missing in the sentence.

Response 3: Thank the reviewer for this content. The sentence was as follows: A total of 18 NRT2 genes belonged to cluster Ⅰ, including five NRT2 genes derived from dicotyledonous Arabidopsis and 13 NRT2 genes from monocotyledonous plants (8, 2, and 3 genes from barley, rice, and maize, respectively). AtNRT2.5 and AtNRT2.7 represented clusters Ⅱ and Ⅲ, containing four and three genes, according to the phylogenetic tree, respectively (Fig 2A). We have revised this in the new manuscript.

Comment 4: Line 256, peaks at ??

Response 4: Thank the reviewer’s for this content. We have revised this as follows: The expression of HvNRT3.3 displayed two peaks at 3 h and 24 h and responded more quickly to high rather than low nitrogen conditions.

Comment 5. Line 261, how much similar are they (% identity)?

Response 5: According to the reviewer’s suggestion. We analyzed the homology identity between HvNRT2.1 and OsNRT2.3. The detail information was as follow: The HvNRT2.1 protein had a similar amino acid length and showed 83.3% identity with OsNRT2.3a, but only 78.4% identity with OsNRT2.3b genes, indicating that HvNRT2.1 plays an important role in long-distance nitrate transport, from the roots to shoots, with low nitrate supply levels in barley. We have added this in the new manuscript (see “Discussion” part).

Comment 6: Line 264, was Arabidopsis ecotype Col-0 used? Should not be in italic.

Response 6: Thank the reviewer for this content. The Arabidopsis thaliana Columbia-0 (Col-0) ecotype was used as in materials. We have revised this in the new manuscript (see “Plant materials” part)

Comment 7: Line 267, how much increments?

Response 7: According to the reviewer’s suggestion, the increments were calculated in the new revised manuscript. The detail information was as follows: 7.9%, 18.5%, and 23.3% increments in the length of siliques, yield per plant and grain number per siliques, respectively, were observed in the transgenic lines compared with Col-0 (Fig 6C, E, and F).

Comment 8. Line 278, two NRT3 from which species?

Response 8: Two NRT3 have been analyzed from rice, Arabidopsis and maize. Thus, we have revised this sentence in the manuscript as follows: a total of 4, 7, and 4 NRT2 genes have been analyzed in rice, Arabidopsis, and maize, respectively, as well as two NRT3 genes.

Comment 9. Line 282, what does “components” mean?

Response 9: For HATS activity in plants, NRT2/NRT3 is a two-component high-affinity nitrate transport system, each of them is part of high-affinity nitrate transport system and interact at the protein level.

Comment 10: Line 307-309, please specify the details in the changes.

Response 10: According to the reviewer’s suggestion, we have added the results in the “HvNRT2/3 gene expression patterns at different developmental stages” and “HvNRT2/3 gene expression analysis under low and high nitrogen levels” parts. 

Comment 11: Line 313-318, state one or two functions in rice so that it can be more evidently related to barley. 

Response 11: According to the reviewer’s suggestion, we have added the discussion in the revised manuscript as follows: In rice, a decreased expression of OsNRT2.3a is related to the accumulation of significantly higher levels of nitrate and total nitrogen in the root and lower levels in the shoot. However, enhancing the expression of OsNRT2.3b improves the growth, yield, and NUE in rice.

Comment 12. Line 335, the statement has been repeated a couple of times in the ms.

Response 12: Thank the reviewer for this content, we have deleted the repeat sentence about this in the revised manuscript.

Comment 13. Line 338, increase of biomass and plant height in Arabidopsis or ?? If so could be rephrased as “In the present study, overexpression of HvNRT2.1 led to high nitrate and nitrogen content, increased biomass and plant height and improved the yield-related traits in Arabidopsis indicating that….

Response 13: Thank the reviewer for this content. We check the result, and adjust the discussion in the new revised manuscript. The detail information as follows: In the present study, the overexpression of HvNRT2.1 led to high nitrate and nitrogen contents and improved the yield-related traits of Arabidopsis. However, both a biomass and plant height increase were not observed in the overexpression plants, indicating that the HvNRT2.1 gene plays a specific role in nitrate and total nitrogen accumulation in plants.

Comment 14. The authors should provide either qualitative or quantitive information regarding the HvNRT2.1 transcript level in transgenic Arabidopsis lines.

Response 14: According to the reviewer’s suggestion. We have added this information in S3 Fig in the reviewed manuscript (see “Morphological characterization of the HvNRT2.1 overexpression plants” part). The detail information as follows: In the transgenic lines, HvNRT2.1 was introduced into the genomic DNA and was expressed at the transcription level in three independent overexpression plants

Comment 15: HvNRT2.1 expression in barley responded to different N levels so why not investigate the growth performance of transgenic Arabidopsis lines under N treatments?

Response 15: We agree with this suggestion. We have plant wide type and transgenic Arabidopsis lines on the MS (without NH4NO3) with different KNO3 level, unfortunately, all plants cannot survive. Thus, the growth performance of wide type and transgenic lines was observed on the normal conditions.

Comment Minor issues

Comment 1: Line 20, rectify the grammar issue

Response 1: Thank the reviewer for this comment. We have corrected the sentence as follow: Nitrogen use efficiency (NUE) is the efficiency with which plants acquire and use nitrogen.

Comment 2. Line 23, in barley “nitrate transport”

Response 2: Thank the reviewer for this content. We have added the word “nitrate transport” in the new manuscript.

Comment 3: Line 25, “chromosomes”

Response 3: Thank the reviewer for this comment. We have replaced the “chromosome” with “chromosomes”.

Comment 4: Line 29, replace “notably” with “furthermore”

Response 4: Thank the reviewer for this comment. We have replaced the “notably” with “furthermore”.

Comment 5: Line 55, should be “they are” but not “it is”

Response 5: Thank the reviewer for this comment. We have replaced the “it is” with “they are”.

Comment 6: Line 61, under low N condition?

Response 6: Nitrate supply was 0.5 mM in the reference (Tang et al., 2012), which was consider as low nitrate supply.

Comment 7: Line 86-89, confusing with the experimental layout. Indeed, how many plants were sampled and used for RNA extraction?

Response 7: According to the reviewer’s suggestion. We have revised this sentence as follow: At 0 h, 1 h, 3 h, 6 h, 12 h, 24 h, 48 h and 72 h after treatment, total roots were harvested from ten plants and immediately frozen in liquid nitrogen for RNA extraction with three biological replicates at each time point.

Comment 8: Line 137, “0.5 cm” would be better.

Response 8: Thank the reviewer for this comment. We have replaced the “mm” with “cm”.

Comment 9: Line 224, according “to”

Response 9: Thank the reviewer for this comment. We have added this word in the revised manuscript.

Comment 10. Line 228, “clustered”

Response 10: Thank the reviewer for this comment. We have a replaced the “cluster” with “clustered”

Comment 11: Line 320, “encode”

Response 11: Thank the reviewer for this comment. The word “encoding” has been replace by “encode”.

---

## [Decision Letter · Decision Letter 1]

3 Apr 2020

PONE-D-19-33678R1

Characterization of the Nitrate Transporter gene family and functional identification of HvNRT2.1 in barley (Hordeum vulgare L.)

PLOS ONE

Dear Dr. Xu,

Thank you for submitting your manuscript to PLOS ONE. After careful consideration, we feel that it has merit but does not fully meet PLOS ONE’s publication criteria as it currently stands. Therefore, we invite you to submit a revised version of the manuscript that addresses the points raised during the review process.

We would appreciate receiving your revised manuscript by May 18 2020 11:59PM. To enhance the reproducibility of your results, we recommend that if applicable you deposit your laboratory protocols in protocols.io, where a protocol can be assigned its own identifier (DOI) such that it can be cited independently in the future. For instructions see: http://journals.plos.org/plosone/s/submission-guidelines#loc-laboratory-protocols

We look forward to receiving your revised manuscript.

Kind regards,

Guoping Zhang

Academic Editor

PLOS ONE

Reviewers' comments:

Reviewer's Responses to Questions

**Comments to the Author**

1. If the authors have adequately addressed your comments raised in a previous round of review and you feel that this manuscript is now acceptable for publication, you may indicate that here to bypass the “Comments to the Author” section, enter your conflict of interest statement in the “Confidential to Editor” section, and submit your "Accept" recommendation.

Reviewer #1: All comments have been addressed

Reviewer #2: (No Response)

2. Is the manuscript technically sound, and do the data support the conclusions?

Reviewer #1: Yes

Reviewer #2: Yes

3. Has the statistical analysis been performed appropriately and rigorously? 

Reviewer #1: Yes

Reviewer #2: Yes

4. Have the authors made all data underlying the findings in their manuscript fully available?

Reviewer #1: Yes

Reviewer #2: Yes

5. Is the manuscript presented in an intelligible fashion and written in standard English?

Reviewer #1: Yes

Reviewer #2: Yes

6. Review Comments to the Author

Reviewer #1: I appreciated that the authors have addressed to all of my concerns. I have no further comments for this manuscript.

Reviewer #2: The revised manuscript has been improved and I have the following comments in regard to typos or grammar errors. In addition, I suggest the authors adding the gene accession numbers or gene IDs such as HORVU.. as the last section of Methods, for a better understanding and traceback with other researches.

Line 37: Replace “serious” with “severe/intense”

Line 48: Replace “shows” with “functions as”

Line 53: Can be rephrased as “There is a lesser number of NRT2 genes than NRT1”

Line 177: “herbicide treatment”

Line 227: Check the 3 distinct clusters

Line 230: It has to be cluster II and III?

Line 330: Replace “with” with “under”

7. PLOS authors have the option to publish the peer review history of their article (what does this mean?). If published, this will include your full peer review and any attached files.

Reviewer #1: Yes: Ping Yang

Reviewer #2: Yes: Yong Han

---

## [Author Response · Author response to Decision Letter 1]

5 Apr 2020

Responses to reviewer 2

Comment 1: The revised manuscript has been improved and I have the following comments in regard to typos or grammar errors. In addition, I suggest the authors adding the gene accession numbers or gene IDs such as HORVU.. as the last section of Methods, for a better understanding and traceback with other researches.

Response 1: Thank the reviewer for this comment. We have listed the gene IDs, chromosome position, coding sequence length, amino acid length, mass and pI in the Table 1 in the revised manuscript.

Comment 2: Line 37: Replace “serious” with “severe/intense”

Response 2: Thank the reviewer for this comment. We have replaced the “serious” with “severe”.

Comment 3: Line 48: Replace “shows” with “functions as”

Response 3: Thank the reviewer for this comment. We have replaced the “shows” with “functions as”.

Comment 4: Line 53: Can be rephrased as “There is a lesser number of NRT2 genes than NRT1”.

Response 4: We agreed with the reviewer’s suggestion, and rephrased as “There is a lesser number of NRT2 genes than NRT1” in the revised manuscript.

Comment 5: Line 177: “herbicide treatment”

Response 5: We agreed with the reviewer’s suggestion. we have added the word “treatment” after “herbicide”

Comment 6: Line 227: Check the 3 distinct clusters

Response 6: Thank the reviewer for this comment. The words“Ⅰ, Ⅱ, and Ⅲ”have not displayed properly in the PDF version. The sentence as follows: According to the phylogenetic tree of the NRT2 and NRT3 proteins in barley, rice, maize, and Arabidopsis, the NRT2 proteins could clearly be divided into three distinct clusters (Ⅰ, Ⅱ, and Ⅲ). 

Comment 7: Line 230: It has to be cluster II and III?

Response 7: Thank the reviewer for this comment. The words“Ⅱ, Ⅲ”have not displayed properly in the PDF version. The sentence as follows: AtNRT2.5 and AtNRT2.7 represented clusters Ⅱ and Ⅲ, containing four and three genes, according to the phylogenetic tree, respectively.

Comment 8: Line 330: Replace “with” with “under”

Response 8: We agreed with the reviewer’s suggestion. We have replaced the “with” with “under”.

---

## [Editor Report · Decision Letter 2]

7 Apr 2020

Characterization of the Nitrate Transporter gene family and functional identification of HvNRT2.1 in barley (Hordeum vulgare L.)

PONE-D-19-33678R2

Dear Dr. Xu,

We are pleased to inform you that your manuscript has been judged scientifically suitable for publication and will be formally accepted for publication once it complies with all outstanding technical requirements.

With kind regards,

Guoping Zhang

Academic Editor

PLOS ONE
---

## [Editor Report · Acceptance letter]

10 Apr 2020

PONE-D-19-33678R2 

Characterization of the Nitrate Transporter gene family and functional identification of HvNRT2.1 in barley (Hordeum vulgare L.) 

Dear Dr. Xu:

I am pleased to inform you that your manuscript has been deemed suitable for publication in PLOS ONE. Congratulations! Your manuscript is now with our production department. 

With kind regards,

on behalf of

Prof Guoping Zhang 

Academic Editor

PLOS ONE